# Intake of Caffeine and Its Association with Physical and Mental Health Status among University Students in Bahrain

**DOI:** 10.3390/foods9040473

**Published:** 2020-04-10

**Authors:** Haitham Jahrami, Mana Al-Mutarid, Peter E. Penson, Mo’ez Al-Islam Faris, Zahra Saif, Layla Hammad

**Affiliations:** 1Ministry of Health, Manama P.O. Box 12, Bahrain; zsaif@health.gov.bh (Z.S.); lhammad@health.gov.bh (L.H.); 2College of Medicine and Medical Sciences, Arabian Gulf University, Manama P.O. Box 26671, Bahrain; mana_1418@icloud.com; 3School of Pharmacy and Biomolecular Sciences, Liverpool John Moores University, Liverpool L3 3AF, UK; p.penson@ljmu.ac.uk; 4Liverpool Centre for Cardiovascular Science, Liverpool L69 3BX, UK; 5Department of Clinical Nutrition and Dietetics, College of Health Sciences/Research Institute of Medical and Health Sciences (RIMHS), University of Sharjah, Sharjah P.O. Box 27272, UAE; mfaris@sharjah.ac.ae

**Keywords:** caffeine, coffee, energy drink, tea, university students

## Abstract

In Western populations, the caffeine intake of young adults has received significant attention in the research literature; our knowledge in other societies remained limited. The objective of this research is to quantify the amount of ingested caffeine and how this is related to measures of physical and mental health in a Bahraini population. A semi-quantitative food frequency questionnaire was used to estimate caffeine intake from coffee, tea, cocoa, soft drinks, energy drinks, chocolates, and over-the-counter medications. Associations between caffeine intake, demographic variables and 25 symptoms measured using the Hopkins Symptoms Checklist-25 were examined. A convenience sample of university students in Bahrain (*n* = 727) was surveyed. Caffeine, in any form, was consumed by 98% of students. Mean daily caffeine consumption was 268 mg/day, with males consuming more than females. Coffee was the main source of caffeine intake, followed by black tea and energy drinks. Participants consuming 400 mg/day or more showed a statistically and significantly twice as high risk for five symptoms, these were: headaches, spells of terror or panic, feeling trapped or caught, worrying too much about things, and having feelings of worthlessness. The prevalence of caffeine intake among university students in Bahrain is high. The overall mean intake of caffeine from all sources by university students was within levels considered to be acceptable by many dietary recommendations. High caffeine intake was associated with an anxiogenic effect in the surveyed students.

## 1. Introduction

Caffeine is the most widely consumed central nervous system stimulant in the world [1] and one of the most extensively studied elements in the human diet [1,2]. Caffeine can be ingested in foods made from natural sources such as coffee, tea, and chocolates [3]. However, synthetic caffeine is often added to food products and beverages to enhance their stimulant properties [3]. Several studies have quantified the ingested caffeine by adults [3,4,5,6]. Estimates suggest that adults consume a daily average of 180–190 mg caffeine [3,4,5,6], which is about two to three cups of coffee. Coffee appeared to be the major source of caffeine, followed by soft drinks and tea [3].

The popularity of caffeine stems from the various subjective benefits that individuals associate with its intake; these include increased attentiveness and alertness, increased work performance, enhanced vigilance, elevated mood and delayed onset of sleep [7]. Different populations use caffeine for various reasons; for example, doctors and surgeons use caffeine to reduce fatigue and increase alertness [8], athletes use caffeine to enhance their physical performance [9], while young adults use caffeine to get more energy, or for the taste, or as part of social gathering or for image enhancement [10]. 

Coffee intake varies worldwide: Nordic countries have the largest intake: people in Finland, Norway, Iceland, Denmark, and Sweden consume an average of 12 kg, 9.9 kg, 9.0 kg, 8.7 kg and 8.2 kg of coffee per capita per year, respectively [11]. Our knowledge of other societies remains limited. 

Several studies showed that consuming a moderate amount of caffeine has a protective effect against cancer [12], diabetes mellitus type 2 [13], inflammatory diseases and pain [14], Parkinson’s and related neurodegenerative diseases [15], cardiovascular disease [16], and stroke [17]. Light to moderate caffeine intake has also been found to be associated with reduced risk of death [18] from all causes, including suicide [19].

However, intake of caffeine in high doses may lead to adverse effects on health [20]. A recent systematic review identified several unwanted symptoms associated with a high daily intake of caffeine; these include palpitations, headache, tremors, anxiety, agitation, restlessness, and sleep problems [21]. 

Research also shows that university students might be at a particularly high risk of adverse effects due to their high intake of caffeine [4]. For example, high caffeine use by university students is associated with sleep problems, particularly poor sleep duration and quality as well as excessive daytime sleepiness [22,23,24]. University students who are trying to control or lose weight are more likely to consume higher amounts of caffeine [25]. Binge alcohol drinking is also associated with the intake of energy drinks and other caffeinated beverages by university students [26]. 

Taking all of the above information collectively, it becomes clear that more research is necessary to study the overall intake of caffeine and its potential cumulative effects on physiology and behavior among populations vulnerable to its negative effects. Given the absence of previous research that focuses on the quantification of caffeine intake, its sources and its impact on health among Bahraini university students; the topic warrants further attention. The Bahraini student population consists mainly of Arab and Muslim individuals, which makes them different from other populations. 

The current study aimed to assess caffeine intake from a wide variety of caffeinated products, including beverages, chocolates, gums and over-the-counter (OTC) medications among a convenience sample of university students in Bahrain. The presence and severity of 25 physical and mental symptoms were assessed using the Hopkins Symptom Checklist-25 (HSCL-25). Associations between caffeine intake and symptomatology and selected socio-demographic variables such as sex, race, income, and anthropometric variables were examined. 

We hypothesized that high caffeine intake would be associated with the cluster of symptoms that define anxiety, such as headache, palpitation, tremors, panic attacks and restlessness. 

## 2. Materials and Methods 

The guidelines of the Strengthening the Reporting of Observational Studies in Epidemiology statement were adopted in planning, implementing, and reporting the study [27]. 

### 2.1. Study Design

The cross-sectional research design was used to assess caffeine intake from a wide variety of caffeinated products, including beverages, chocolates, gums and OTC medications among a convenience sample of university students in Bahrain. 

### 2.2. Setting and Participants

The study was conducted using an online survey in December 2019. A convenience sample of university students was recruited through an online information weblink circulated using the social network platform WhatsApp^®^. A link to the questionnaire in an Arabic language Google Forms format was initially posted on several WhatsApp^®^ chat groups of students in six universities: Arabian Gulf University, University of Bahrain, Applied Science University of Bahrain, Ahlia University, Bahrain Polytechnic, and AMA International University of Bahrain. When the students clicked on the link, they were taken to the electronic Google Form. Google Form saves each completely filled questionnaire in the investigator’s Google drive. Upon completing the questionnaire, the students were asked to forward the survey link to their WhatsApp^®^ study groups/siblings/partners. All the completed forms were available to view on the drive, which was password-protected and could be downloaded when needed for analysis. Only students who were not enrolled in any program at a university located in Bahrain or those who were not willing to participate and providing informed consent were excluded. Participants were able to answer the questions within their own time frame, enabling them to have privacy or choice of space. 

Using a margin of error of 5% (alpha error), a confidence level of 95% and a response rate of 50% from a given population of approximately 40,000 students in Bahrain, we estimated that 380 would be the minimum viable sample size for sufficiently powered analyses. The endpoint used to power the analysis was assuming that 15% or more of the students would consume high amounts of caffeine (≥400 mg/day).

### 2.3. Tools and Techniques

An Arabic language, self-administered questionnaire was used to collect the data. The questionnaire consisted of structured, closed-ended questions. There were no open-ended or continuing questions, making the questionnaire simple and quick to answer; the investigator estimated that it would take each participant around 7 to 10 min to complete their form based on a pilot test activity. The questionnaire was divided into three domains; socio-demographics and anthropometrics, daily caffeine intake, and the HSCL-25 [28]. 

### 2.4. Variables

As described above, the survey collected data on a number of socio-demographic and anthropometrics. These included: sex, marital status, income, race, university major, tuition payment plan, academic year, general health, age and self-reported anthropometrics (weight and height). Self- reported measurement has been found to be both valid and reliable when compared to measurements taken directly by researchers [29,30,31]. Body mass index (BMI) was calculated using reported weight and height, dividing the weight in kilograms on the squared height in meters. Individuals with BMI < 18.5 were classified as underweight, individuals with BMI ≥ 18.5 and <24.9 were classified as being in the normal weight range, while a BMI ≥ 25.0 and <29.9 was considered as overweight and individuals with BMI ≥ 30.0 were classified as obese. 

The self-report survey instrument included detailed semi quantitative food frequency questionnaire questions on types of caffeine-containing products consumed, and the serving size and the frequency of intake. A diverse variety of 38 caffeine-containing items were included as a response option in the caffeine intake section. The caffeine-containing items were: coffee, decaf coffee, concentrated coffee including Arabic coffee and espresso, black tea, green tea, cocoa, energy drinks, soft drinks (sodas/fizzy drinks), chocolates and gums, and finally OTC caffeine-containing analgesics. Based on the information reported on product type and serving size, the daily caffeine intake was calculated using data on the amount of caffeine in each specific product. Sources of caffeine content in specific products included: United States Department of Agriculture Nutrient Database release 28 product labels, product websites, and other reliable online sources of caffeine content. 

The Arabic validated HSCL-25 is well-known and widely used to screen for psychological distress, anxiety, and depression [28]. It consists of 25 symptoms answered on a Likert-like four categories of response (“Not at all”, “A little bit”, “Quite a bit”, “Extremely”, rated 1 to 4, respectively). Three scores are calculated from the HSCL-25: the total score is the average of all 25 items, while the depression score is the average of 15 items, and the anxiety score is the average of 10 items. It has been consistently shown among several populations that the total score is highly correlated with the severe emotional distress of unspecified diagnosis, the depression score is correlated with major depression and the anxiety score is correlated with anxiety, as defined by the Diagnostic and Statistical Manual of the American Psychiatric Association, 4th Edition. The 25 individual symptoms of the HSCL-25 and the corresponding three scores of psychological distress, anxiety and depression were classified to binary for subsequent analyses. For the individual 25 symptoms, a response of “Not at all” was considered normal, while a response of “A little bit” or “Quite a bit” or “Extremely” were considered symptomatic. For psychological distress, anxiety and depression, a cut-off of 1.75 was used to differentiate normal from abnormal or pathological scores. 

### 2.5. Ethical Consideration

The Research Ethics Committee (REC) of the Arabian Gulf University approved the research (E15-PI-12/19), and data collection was started following the approval. Electronic informed consent was sought and obtained from the participants. Participation was voluntary; no monetary or non-monetary incentives were given and the participant was permitted to withdraw at any time. 

### 2.6. Statistical Analysis

The data were analyzed using STATA 16 (Version 16, 2019, College Station, TX: StataCorp LLC) [32]. Descriptive statistics were used to summarize participants’ socio-demographic and anthropometric characteristics; the arithmetic mean and standard deviation (SD) were reported for continuous variables, and count and percentage were reported for categorical variables. To compare the two groups, the independent sample t-test was used for continuous variables and Chi-square was used for categorical variables. 

The U.S. Food and Drug Administration has cited 400 milligrams per day as a safe dose of caffeine [33]. Thus, to examine the association between high caffeine intake (≥400 mg/day) and the symptoms of the HSCL-25, logistic regression was performed, and odds ratio (OR) and 95% confidence intervals (95% CI) were computed, and significance was considered at *p*-value < 0.05. In the regression model, the independent variable was high caffeine intake (≥400 mg/day) or normal caffeine intake (<400 mg/day). The dependent variable was absence of symptoms (not at all) or the presence of symptoms (a little bit, quite a bit, extremely) for the individual scores pf the symptoms of the HSCL-25; or a cut off of 1.75 for psychological distress, anxiety or depression scores. 

## 3. Results

Descriptive socio-demographic and anthropometric characteristics of the study participants are presented in Table 1. The majority of the participants were female (approx. 63%), single (approx. 92%), Arab race (approx. 97%), with a monthly income of ≥$500 (approx. 68%). The mean age was 20.72 ± 1.9 and the mean BMI of 24.0 ± 5.46. 

The daily intake of caffeine of the study participants is presented in Table 2. The vast majority (approx. 98%) of the participants reported regular daily intake caffeine products. Approximately 76% of the participants consumed at least one cup of regular coffee per day, 20% consumed at least one cup of decaffeinated coffee per day, 55% consumed at least one shot of concentrated coffee per day, 72% consumed at least one cup of black tea per day, 20% consumed at least one cup of green tea per day, 20% consumed at least one cup of cocoa per day, 22% consumed at least one can of energy drink per day, 55% consumed at least one can of soft drink per day, 70% consumed at least one bar of chocolate per day, and 34% consumed at least one tablet of OTC pain remedies containing caffeine per day.

The mean daily caffeine intake was estimated to be 268 mg/day from all sources. The main sources of caffeine: regular coffee, black tea, concentrated coffee, and energy drinks with 133 mg/day, 81 mg/day, 61 mg/day, and 56 mg/day were obtained from these products per person; respectively.

Mean caffeine intake from all sources was higher for males (306 mg/day) than for females (246 mg/day) (*p* = 0.01). There were no statistically significant differences between males and females in the amount of caffeine consumed from regular coffee, decaf coffee, concentrated coffee, green tea, cocoa, energy drinks, and chocolate and gums. Males consumed more caffeine from black tea (*p* < 0.001) and soft drinks (*p* < 0.001) compared to females. Females consumed more caffeine from OTC medications (*p* < 0.001) compared to males.

About 18% of the study participants consumed 400 mg/day or more according to Table 2. Only four participants (0.5%) consumed 2000 mg/day or more.

Table 3 provides the distribution of the symptoms listed in the HSCL-25 of the study participants. The most prevalent symptoms were “Nervousness or shakiness inside”, “Feeling no interest in things”, “Blaming yourself for things”, “Headaches”, “Feeling everything is an effort”, “Feeling blue”, “Feeling restless, can’t sit still”, “Difficulty falling asleep, staying asleep”, “Feeling low in energy--slowed down”, and “Feeling fearful”. Scores of the HSCL-25 show that approx. 47% experienced anxiety disorders and approximately 9% experienced depressive disorders. The global score indicates that just more than half (55%) experienced psychological distress.

Table 4 provides the strength of association between consuming a high amount of caffeine (>400 mg/day) and the symptoms of the HSCL-25. Participants consuming 400 mg/day or more showed statistically significantly difference for five symptoms, these were: headaches (OR = 1.84, *p* = 0.02, 95% CI = 1.11–3.03), spells of terror or panic (OR = 1.82, *p* = 0.001, 95% CI = 1.21–2.75), feeling trapped or caught (OR = 2.05, *p* = 0.001, 95% CI = 1.32–3.19), worrying too much about things (OR = 1.72, *p* = 0.01, 95% CI = 1.16–2.53), feelings of worthlessness (OR = 1.57, *p* = 0.03, 95% CI = 1.06–1.06), anxiety score (OR = 1.80, *p* = 0.001, 95% CI = 1.22–2.64), and psychological distress score (OR = 1.79, *p* = 0.001, 95% CI = 1.20–2.66).

## 4. Discussion

The present study aimed to quantify caffeine intake by university students using data from a convenience sample of over 700 students from different universities in Bahrain. The study is based on a detailed survey that examined the frequency and quantity of intake of an extensive range of commonly available caffeinated products. The study also used the HSCL-25 to examine the association between high caffeine intake and mental health.

About 98% of students reported regular daily intake of caffeine, with the majority consuming coffee, tea, and soft drinks. The total mean caffeine intake was 268 mg/day. Our results are slightly higher than the published estimates of the intake of caffeine by U.S. and Dutch university students, who consume an average of 159 mg/day and 144 mg/day, respectively [6,34]. One possible explanation for the higher intake of caffeine in our study population is the fact that caffeinated beverages are the most commonly available drink products for university students and young adults in Arab and Muslim countries. This proposition is supported by the findings of caffeine intake among university students in a neighboring country, the United Arab Emirates (UAE), where caffeine intake was estimated to be approximately 250 mg/day [35]. In the Arab region, caffeinated beverages, especially tea and coffee, are an essential part of hospitability in every social event, thus, their intake is influenced by social norms.

Similar to U.S., Dutch and UAE students, Bahraini students surveyed consumed most of their caffeine from coffee (133 mg/day) and tea (81 mg/day). The daily intake of caffeine from energy drinks in Bahraini students was 56 mg/day, which is similar to results from the U.S. [6] Consistent with previous research, our study suggests that caffeinated beverages make the largest contribution to the total caffeine intake per day amongst all the examined sources of caffeine [3,34].

The overall mean intake of caffeine from all dietary and non-dietary sources by university students was within levels considered to be acceptable by many dietary recommendations. About one-fifth of the students consumed more caffeine (over 400 mg/day) than is advised [7,36]. Levels of unsafe or maximum caffeine intake remain debatable, due to limited safety data. Our research shows that high caffeine use is associated with the following symptomatology: headaches, spells of terror or panic, feeling trapped or caught, worrying too much about things, anxiety and psychological distress. The present study is the first and largest to examine the association of daily caffeine intake from caffeine-containing products with symptoms among university students in Bahrain.

Because this is an observational study, we cannot demonstrate a causal link between caffeine intake and present pointed out anxiety-related symptoms reported by students in this study. The pharmacology of caffeine and related methylxanthines is complex, as they modulate a variety of biological targets. Nevertheless, clinical and experimental investigations of caffeine pharmacology provide biological plausibility to some of these effects, as outlined in a recent comprehensive review [37]. At physiologically relevant concentrations, the predominant pharmacological effect is as a purinoceptor antagonist. Caffeine acts at a number of adenosine receptors, including A1 and A2A. The A1A predominates in the brain, and the activation of A1A has been associated with anxiolytic effects [38]. Indeed, positive allosteric modulators of A1A have been proposed as a therapeutic strategy for anxiety [38]. Based upon laboratory observations, it is often claimed that caffeine is a phosphodiesterase inhibitor, however, this action is unlikely to occur, at any but the very highest concentrations in vivo, and the relevance to anxiety is unclear [37].

A recent comprehensive systematic review of the possible adverse effects of caffeine on the cardiovascular system, bone status, reproductive health, and development, as well as behavior, concluded that, for adults there was no evidence that a caffeine intake of up to 400 mg/day posed any risk of adverse effects [21].

The strengths of this study are numerous. Caffeine intake among university students was quantified by consideration of many possible sources, rather than only from caffeinated beverages. We included OTC medications, chocolate and gums. The examination of the association between caffeine intake and a list of common mental/psychological symptoms related to mental health is a novel contribution to the literature.

There are several limitations to this study. The major limitation is that the obtained data were self-reported and various types of biases, e.g., recall bias, are becoming a challenge. The study is based on a self-selecting convenience sample that is unlikely to be representative of the entire university student population in Bahrain. Because this is an observational study, rather than a randomized trial, we can only study associations between variables; we cannot demonstrate causality. Although we adjusted the odds ratios in our logistic regression model to account for demographic data, it is very likely that residual confounding exists.

## 5. Conclusions

The majority of university students consume caffeine on a daily basis; coffee is the main source of ingested caffeine. The mean intake was 268 mg/day, with males consuming more caffeine than females. High caffeine intake in this population was associated with headaches, anxiety, and psychological distress.

## Figures and Tables

**Table 1 foods-09-00473-t001:** Descriptive sociodemographic and anthropometric characteristics of the study participants.

Variable	Participants, n = 727 n(%)/Arithmetic Mean ± SD
Sex	
Male	269 (37%)
Female	458 (63%)
Marital status	
Not married	669 (92.02%)
Married	58 (7.98%)
Income/month	
≥$500	495 (68.09%)
$500–$1000	101 (13.89%)
+$1000	131 (18.02%)
Race	
Arab	705 (96.97%)
Non-Arab	22 (3.03%)
University major	
Medical and Health Sciences	310 (42.64%)
Education	22 (3.03%)
Engineering	78 (10.73%)
Law	69 (9.49%)
Business Administration	116 (15.96%)
Computer Sciences	25 (3.44%)
Other specialties	107 (14.72%)
Tuition payment plan	
Personal	280 (38.51%)
Scholarship	358 (49.245)
Other Funds	89 (12.24%)
Academic year	
First	242 (33.29%)
Second	156 (21.46%)
Third	101 (13.89%)
Fourth	104 (14.31%)
Fifth	49 (6.74%)
Sixth	75 (10.32%)
Age (year)	20.72 ± 1.9
Height (cm)	163.91 ± 9.22
Weight (kg)	64.73 ± 16.53
Body mass index, BMI (kg/m^2^)	24.0 ± 5.46

**Table 2 foods-09-00473-t002:** Daily caffeine intake by the study participants.

Part 1: Daily Intake Frequency of Caffeine by Source	
Variable *	Participants, n = 727
Intake frequency of regular coffee	
None	174 (23.93%)
1–2 Units	439 (60.39%)
3–4 Units	94 (12.93%)
≥5 Units	20 (2.76%)
Intake frequency of decaf coffee	
None	589 (81.02%)
1–2 Units	107 (14.72%)
3–4 Units	21 (2.89%)
≥5 Units	10 (1.38%)
Intake frequency of concentrated coffee	
None	326 (44.84%)
1–2 Units	275 (37.83%)
3–4 Units	89 (12.24%)
≥5 Units	37 (5.09%)
Intake frequency of black tea	
None	205 (28.2%)
1–2 Units	371 (51.03%)
3–4 Units	95 (13.07%)
≥5 Units	56 (7.70%)
Intake frequency of green tea	
None	597 (82.12%)
1–2 Units	109 (14.99%)
3–4 Units	16 (2.20%)
≥5 Units	5 (0.69%)
Intake frequency of cocoa	
None	583 (80.19%)
1–2 Units	132 (18.16%)
3–4 Units	6 (0.83%)
≥5 Units	6 (0.83%)
Intake frequency of energy drinks	
None	562 (77.3%)
1–2 Units	144 (19.81%)
3–4 Units	11 (1.51%)
≥5 Units	10 (1.38%)
Intake frequency of soft drinks	
None	326 (44.84%)
1–2 Units	346 (47.59%)
3–4 Units	42 (5.78%)
≥5 Units	13 (1.79%)
Intake frequency of chocolate and gums	
None	212 (29.16%)
1–2 Units	431 (59.28%)
3–4 Units	65 (8.94%)
≥5 Units	19 (2.34%)
Intake frequency of over the counter medications (tablet)	
None	482 (66.3%)
1–2 Units	172 (23.66%)
3–4 Units	48 (6.60%)
≥5 Units	11 (3.44%)
**Part 2: Mean Daily Intake of Caffeine by Source**	
Regular coffee mg/day	133.29 ± 130.93
Decaf coffee mg/day	1.15 ± 3.21
Concentrated coffee mg/day	61.14 ± 85.24
Black tea mg/day	80.88 ± 94.03
Green tea mg/day	10.69 ± 32.56
Cocoa mg/day	4.5 ± 13.94
Energy drinks mg/day	56.53 ± 159.1
Soft drinks mg/day	27.24 ± 38.16
Chocolate and gums mg/day	18.24 ± 21.85
Over the counter medications mg/day	7.66 ± 15.61
All sources mg/day	268.03 ± 319.83
**Part 3: Prevalence of High Intake of Caffeine**	
Caffeine ≥400 mg/day	130 (17.88%)
Caffeine ≥500 mg/day	82 (11.28%)
Caffeine ≥600 mg/day	51 (7.02%)
Caffeine ≥700 mg/day	32 (4.40%)
Caffeine ≥800 mg/day	20 (2.75%)
Caffeine ≥900 mg/day	17 (2.34%)
Caffeine ≥1000 mg/day	15 (2.06%)
Caffeine ≥2000 mg/day	4 (0.55%)

* Units are Standard Cup = 240 milliliters (8 US fluid ounces), Shot cup = 30 milliliters (1 US fluid ounces), Soft drink can =330 milliliters (11 US fluid ounces), Energy drink can =250 milliliters (8 US fluid ounces), Bar = 57 g (2 ounces).

**Table 3 foods-09-00473-t003:** Distribution of Hopkins Symptom Checklist—25 items (HSCL-25) of the study participants.

Symptom of HSCL-25	Healthy	Symptomatic *		
	Not at all (1)	A little bit (2)	Quite a bit (3)	Extremely (4)
Suddenly scared for no reason	322 (44.29%)	319 (43.88%)	63 (8.67%)	23 (3.16%)
Feeling fearful	251 (34.53%)	385 (52.96%)	69 (9.49%)	22 (3.03%)
Faintness, dizziness, or weakness	393 (54.06%)	251 (34.53%)	70 (9.63 %)	13 (1.79%)
Nervousness or shakiness inside	131 (18.02%)	316 (43.47%)	202 (27.79%)	78 (10.73%)
Heart pounding or racing	277 (38.1%)	312 (42.92%)	90 (12.38%)	48 (6.60%)
Trembling (Tremors)	420 (57.77%)	221 (30.40%)	67 (9.22%)	19 (2.61%)
Feeling tense or keyed up	348 (47.87 %)	239 (32.87%)	92 (12.65 %)	48 (6.60 %)
Headaches	177 (24.35%)	366 (50.34 %)	129 (17.74%)	55 (7.57%)
Spells of terror or panic	552 (75.93%)	122 (16.78%)	34 (4.68%)	19 (2.61%)
Feeling restless, can’t sit still	243 (33.43%)	308 (42.37%)	115 (15.82%)	61 (8.39%)
Feeling low in energy—slowed down	245 (33.7 %)	297 (40.85%)	120 (16.51%)	65 (8.94%)
Blaming yourself for things	176 (24.21%)	252 (34.66%)	178 (24.48%)	121 (16.64%)
Crying easily	266 (36.59%)	233 (32.05%)	122 (16.78%)	106 (14.58%)
Loss of sexual interest or pleasure	480 (66.02%)	139 (19.12%)	57 (7.84%)	51 (7.02%)
Poor appetite	310 (42.64%)	313 (43.05%)	78 (10.73%)	26 (3.58%)
Difficulty falling asleep, staying asleep	244 (33.56%)	256 (35.21%)	143 (19.67%)	84 (11.55%)
Feeling hopeless about the future	291 (40.03%)	246 (33.84%)	108 (14.86%)	82 (11.28%)
Feeling blue	239 (32.87%)	304 (41.82%)	105 (14.44%)	79 (10.87%)
Feeling lonely	284 (39.06%)	235 (32.32%)	111 (15.27%)	97 (13.34%)
Feeling trapped or caught	597 (82.12%)	70 (9.63%)	24 (3.30%)	36 (4.95%)
Worrying too much about things	359 (49.38%)	257 (35.35%)	74 (10.18%)	37 (5.09%)
Feeling no interest in things	135 (18.57%)	250 (34.39%)	189 (26%)	153 (21.05%)
Thoughts of ending your life	258 (35.49%)	288 (39.61%)	113 (15.54%)	68 (9.35%)
Feeling everything is an effort	224 (30.81%)	295 (40.58%)	125 (17.19%)	83 (11.42 %)
Feelings of worthlessness	322 (44.29%)	224 (30.81%)	81 (11.14%)	100 (13.76%)
**Scoring of (HSCL-25)**				
Prevalence of anxiety **	344 (47.32%)			
Prevalence of depression ***	65 (8.94%)			
Prevalence of Psychological Distress ****	404 (55.57%)			

* Symptoms distribution; A little bit = mild symptoms, Quite a bit = moderate symptoms, Extremely = severe symptoms; ** Defined by items 1–10, (∑1–10/10) and a score ≥1.75 is considered abnormal; *** Defined by items 11–25, (∑11–25/15) and a score ≥1.75 is considered abnormal; **** Defined by items 1–25, (∑1–25/25) and a score ≥1.75 is considered abnormal.

**Table 4 foods-09-00473-t004:** The association between high caffeine intake (>400 mg/day) and symptoms of the study participants.

Symptom	OR **	*p*-Value	95% Confidence Interval	
Suddenly scared for no reason	1.15	0.49	0.78	1.68
Feeling fearful	1.13	0.56	0.75	1.69
Faintness, dizziness, or weakness	0.90	0.60	0.62	1.32
Nervousness or shakiness inside	1.57	0.11	0.91	2.72
Heart pounding or racing	0.91	0.62	0.62	1.34
Trembling (Tremors)	1.17	0.42	0.80	1.71
Feeling tense or keyed up	1.17	0.41	0.80	1.72
Headaches	1.84	0.02 *	1.11	3.03
Spells of terror or panic	1.82	0.001 *	1.21	2.75
Feeling restless, can’t sit still	1.27	0.26	0.84	1.92
Feeling low in energy—slowed down	0.95	0.81	0.64	1.42
Blaming yourself for things	0.88	0.57	0.57	1.36
Crying easily	0.72	0.09	0.49	1.05
Loss of sexual interest or pleasure	1.17	0.43	0.79	1.74
Poor appetite	0.94	0.76	0.64	1.38
Difficulty falling asleep, staying asleep	1.03	0.90	0.69	1.54
Feeling hopeless about the future	1.27	0.23	0.86	1.89
Feeling blue	0.95	0.80	0.63	1.42
Feeling lonely	1.21	0.34	0.82	1.80
Feeling trapped or caught	2.05	0.001 *	1.32	3.19
Worrying too much about things	1.72	0.01 *	1.16	2.53
Feeling no interest in things	0.75	0.23	0.47	1.19
Thoughts of ending your life	1.19	0.40	0.79	1.78
Feeling everything is an effort	1.00	0.99	0.66	1.51
Feelings of worthlessness	1.57	0.03 *	1.06	2.33
Anxiety score	1.80	0.001 *	1.22	2.64
Depression score	1.29	0.42	0.69	2.41
Psychological distress score	1.79	0.001 *	1.20	2.66

* Significant at 0.05; ** Odds ratio, adjusted for age, sex and BMI.

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
