# Peer review of "Intake of Caffeine and Its Association with Physical and Mental Health Status among University Students in Bahrain"

_foods, 2020, doi:10.3390/foods9040473_

Round 1

Reviewer 1 Report

This paper by Jahrami and collaborators examined the effect of intake of caffeine on physical and mental health status.

Overall, the manuscript is well examined and written.

Therefore, this manuscript should be accepted.

Author Response

Dear Reviewer #1

We would like to thank the editor and the expert reviewers for their detailed comments and suggestions for the manuscript. We trust that we have accommodated all of the comments in our revision as per the attached comments/response below. Changes were made in Track Changes for ease of tracking.

Comments and Suggestions for Authors

Author Responses

This paper by Jahrami and collaborators examined the effect of intake of caffeine on physical and mental health status.

Overall, the manuscript is well examined and written.

Therefore, this manuscript should be accepted.

Thank you so much.

Best wishes,

Reviewer 2 Report

Thank you for allowing me to read this manuscript. I found this to be an interesting and well-written study, and as such have only a few minor comments that the authors may wish to consider.

Should the title reflect the Bahraini population? This could be helpful as it is a specific population.

It would be useful to highlight in the introduction why the Bahraini student population would be different from other populations? This would help to strengthen the rationale for the study.

Line 62: Include a definition of 'high caffeine' e.g. Buhler et al. (2014) caffeine consumption questionnaire defines low habitual caffeine consumption defined as <300 mg·day-1 and >300 mg·day-1 defined as high.

Line 180: Reference and / or rationale for 400 mg·day-1 being defined as high.

Tables: Consider reporting the percentages to one decimal place, or as an integer.

I hope that the authors find these comments helpful and in the constructive manner they were intended. 

Author Response

Dear Reviewer #2

We would like to thank the editor and the expert reviewers for their detailed comments and suggestions for the manuscript. We trust that we have accommodated all of the comments in our revision as per the attached comments/response below. Changes were made in Track Changes for ease of tracking.

Comments and Suggestions for Authors

Author Responses

Thank you for allowing me to read this manuscript. I found this to be an interesting and well-written study, and as such have only a few minor comments that the authors may wish to consider.

Thank you so much.

Should the title reflect the Bahraini population? This could be helpful as it is a specific population.

Done title reads now as “Intake of caffeine and its association with physical and mental health status among university students in Bahrain”

It would be useful to highlight in the introduction why the Bahraini student population would be different from other populations? This would help to strengthen the rationale for the study.

Done Ln 77-79

The Bahraini student population consists mainly from Arab, Muslim individuals which makes them different from other populations.

Line 62: Include a definition of 'high caffeine' e.g. Buhler et al. (2014) caffeine consumption questionnaire defines low habitual caffeine consumption defined as <300 mg·day-1 and >300 mg·day-1 defined as high.

Done.

Bühler E, Lachenmeier DW, Schlegel K, Winkler G. Development of a tool to assess the caffeine intake among teenagers and young adults. Ernahrungs Umschau. 2014;61(4):58-63.

Line 180: Reference and / or rationale for 400 mg·day-1 being defined as high.

Done.

The U.S. Food and Drug Administration has cited 400 milligrams per day as safe dose of caffeine[33].

Tables: Consider reporting the percentages to one decimal place, or as an integer.

This is great idea to simplify the results. However, we would like to retain the current presentation because: a)it allows readers to compare with other similar research, b)for future meta-analyses the current approach is more robust.

I hope that the authors find these comments helpful and in the constructive manner they were intended. 

Your support, feedback and suggestions made the manuscript much better. Thank you.

Best wishes,

Reviewer 3 Report

The paper is generally well written and topical.

L28 with males consuming more than females ie delete were

Intro L 48 associate

L54 lowercase p for people

L 56 remains

L 114 amounts

L132 not on by

L 192 intake of caffeine

L197 soft drink

L 207 delete were

L212 coffee, decaf  ie delete ( insert ,

L 248 convenience spelt wrong

L253 delete were

L283 required editing

L 285 needs full stop at end of sentence

L 300 Just list the strengths don’t say numerous

L301 all possible sources is incorrect- what about foods?

L306 rephrase becoming a challenge

L315 delete were

L316 Capital H for High

Author Response

Dear Reviewer #3

We would like to thank the editor and the expert reviewers for their detailed comments and suggestions for the manuscript. We trust that we have accommodated all of the comments in our revision as per the attached comments/response below. Changes were made in Track Changes for ease of tracking.

Because of added text some line numbers has been changed.

Comments and Suggestions for Authors

Author Responses

The paper is generally well written and topical.

Thank you so much.

L28 with males consuming more than females ie delete were

Done.

Intro L 48 associate

Done.

L54 lowercase p for people

Done.

L 56 remains

Done.

L 114 amounts

Done.

L132 not on by

Done.

L 192 intake of caffeine

Done.

L197 soft drink

Done

L 207 delete were

Done

L212 coffee, decaf  ie delete ( insert ,

Done.

L 248 convenience spelt wrong

Done.

L253 delete were

Done.

L283 required editing

Done.

L 285 needs full stop at end of sentence

Done.

L 300 Just list the strengths don’t say numerous

Done.

L301 all possible sources is incorrect- what about foods?

Changed to many, because we included OTC medications and chocolate and gums.

L306 rephrase becoming a challenge

L315 delete were

L316 Capital H for High

Best wishes,
